# A fuzzy description logic based IoT framework: Formal verification and end user programming

**Miguel Pérez-Gaspar**[1], **Javier Gomez**[1], **Everardo Bárcenas**[2]*, **Francisco Garcia**[1]

**1** Department of Telecommunications, National Autonomous University of Mexico, Mexico City, Mexico,
**2** Department of Computer Engineering, National Autonomous University of Mexico, Mexico City, Mexico

☯ These authors contributed equally to this work.
* ebarcenas@unam.mx

**Data Availability Statement:** Data for experiments: https://kaggle.com/datasets/e9899e7c8b983b13c584b3e7a015b6b7e6f698da6fae8016e5ee21ff3d1086de.

## Abstract

The Internet of Things (IoT) has become one of the most popular technologies in recent years. Advances in computing capabilities, hardware accessibility, and wireless connectivity make possible communication between people, processes, and devices for all kinds of applications and industries. However, the deployment of this technology is confined almost entirely to tech companies, leaving end users with only access to specific functionalities. This paper presents a framework that allows users with no technical knowledge to build their own IoT applications according to their needs. To this end, a framework consisting of two building blocks is presented. A friendly interface block lets users tell the system what to do using simple operating rules such as "if the temperature is cold, turn on the heater." On the other hand, a fuzzy logic reasoner block built by experts translates the ambiguity of human language to specific actions to the actuators, such as "call the police." The proposed system can also detect and inform the user if the inserted rules have inconsistencies in real time. Moreover, a formal model is introduced, based on fuzzy description logic, for the consistency of IoT systems. Finally, this paper presents various experiments using a fuzzy logic reasoner to show the viability of the proposed framework using a smart-home IoT security system as an example.

## Introduction

The Internet of Things (IoT) makes possible communications between people and objects by taking advantage of computing capabilities and hardware accessibility for all types of applications. As a general definition, IoT can be described as an interconnection of many objects through a network that continuously generates information about the physical world. These objects can communicate and be controlled by various agents (other systems or people) to interact and take control of the physical world to manage many services of daily use [1].

IoT devices can generally be classified into controller boards with microprocessors or micro-controllers, sensors that sense data from the physical world, and actuators that connect to controllers and communication modules. However, the development and deployment of

**Funding:** JG, EB, FG: IA104122, IA105420, IA102822; UNAM-PAPIIT program. JG: 0320403; Ciencia de Frontera CONAHCyT. MPG: Estancia-Posdoctoral; UNAM-DGAPA program. The funders had no role in study design, data collection and analysis, decision to publish, or preparation of the manuscript.

IoT applications are confined almost entirely to tech companies, leaving end users with only access to specific functionalities. This approach presents a fundamental problem since it is only possible for the IoT provider to anticipate some of the user's needs. Unfortunately, it will take a long time before changes are made to an IoT system to fulfill a user's specific needs. In this work, we argue that for IoT systems to close the gap between users and IoT technology, a different approach is needed where end users have the means to build their IoT systems according to their specific needs. However, for this end, the interface should be user-friendly, using day-to-day instructions as input, such as "If these conditions happened, then do this or that." Nevertheless, the system should allow users to express complex system behaviors while at the same time verifying that no inconsistencies appear when additional rules are added.

This work proposes that Fuzzy logic (FL) [2] is ideally suited to become the interface between people and objects in IoT systems, modeling logical reasoning with vague or ambiguous statements such as "The temperature is hot (cold or mild)." This logic refers to a family of many-valued logic in which truth values are interpreted as degrees of truth. The truth value of a logically complex proposition such as "Carl is tall, and Chris is rich" is determined by the truth values of its constituents. In other words, truth functions impose on classical logic. This type of logic arises from the need to use daily life statements whose natural language adjectives are used to qualify.

Fuzzy logic has been applied in various ways in sensor networks and IoT, such as energy savings, packet routing, location, and human-sensor interface, among other applications [3–6]. An advantage of fuzzy logic over traditional logic is that the former can reach precise conclusions based on vague, imprecise, noisy, or non-existent arguments common in IoT systems since messages are often lost, collected information is imprecise, or the instructions are vague [7]. Moreover, many IoT applications require human intervention where the user might provide ambiguous inputs to the system, such as higher, smaller, or bigger, that an actuator cannot read directly.

Inconsistent information is recurrent in IoT systems due to many factors, including errors produced by data entry operators, data arriving from multiple sensors, and instructions contradicting each other [8, 9]. Since Fuzzy-IoT systems can only make an action based on the collected information at a given time, it is relevant to verify the consistency of the entire system as soon as new instructions are added to the system; otherwise, the system might present errors when applying these actions. For instance, Let us consider two rules for the activation of warning alarms: "Activate a medium warning alarm when low motion, light, and sound are detected" and "Activate a low warning alarm when high motion, low light, and low sound are detected." These rules exhibit inconsistency since the second rule implies that a low warning alarm should be triggered by high motion, which contradicts the expectation that a medium or high warning alarm should be activated given the potential presence of a stranger inside the house. Thus, even when some smart IoT systems can enhance daily life, their proper operation could be compromised if the rules inserted into the system are inconsistent. To solve this problem, this paper proposes a novel framework that allows users to interact with the IoT systems using simple instructions and vague language while verifying the system's overall consistency. For example, a simple rule can be:

$$\textit{If it is really hot, then turn on the air conditioning high}. \tag{1}$$

While a second rule can be:

$$\textit{No matter what, never turn on the air conditioning}. \tag{2}$$

Fig 1 illustrates the flow diagram of the proposed framework. First, a technician (or even the user) needs to place and connect various sensors and actuators that compose the IoT

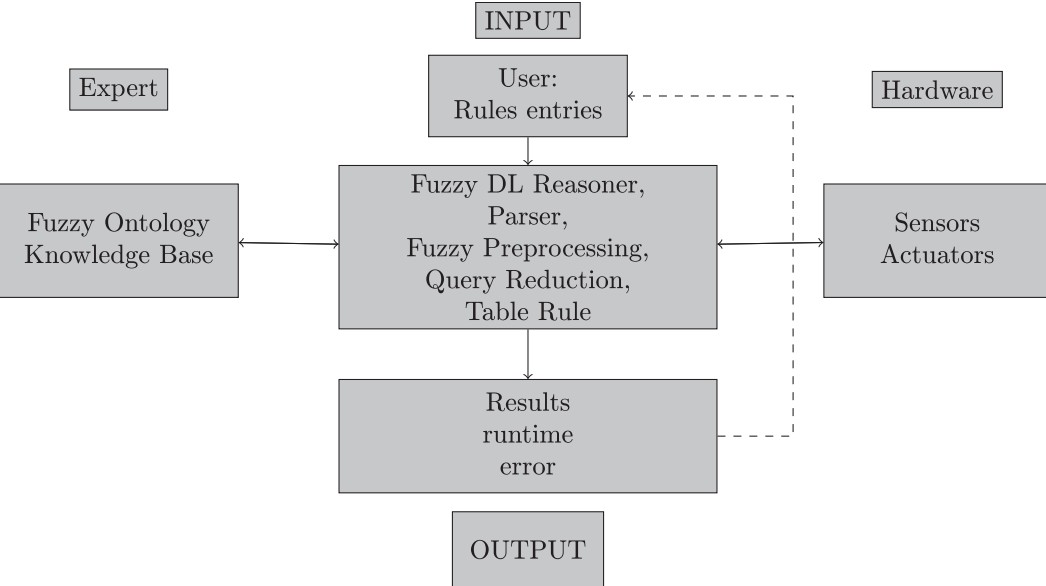

**Fig 1. The architecture of fuzzyDL reasoner.**

hardware on which the IoT system operates. Some important contextual concepts may be pre-programmed onto the system, such as *really hot* and *air conditioning* set to high. Nevertheless, users may modify these concepts later on. The input is given by simple everyday rules such as instructions 1 and 2, then a fuzzy reasoner will process the user's input. If the last instruction contradicts another previous rule, the system will alert the user accordingly. Once the system is programmed with non-contradictory (consistent) instructions, it routinely performs its corresponding tasks (e.g., turn on the heater, set the alarm, etc).

The structure of this work is organized as follows: In the first section, we discuss previous research to provide context for our study. Then, in the IoT Systems section, we define the system and introduce the concept of consistency. Next, in the Fuzzy Description Logic Verification section, we explain the basics of fuzzy logic and present a result that shows how consistency relates to both the IoT system and fuzzy description logic. Ultimately, in the Fuzzy Control for an IoT Security System section, we outline the following steps: providing context, setting system rules, and putting the system into action. Finally, we present the program's syntax and share the experiments we conducted to support and illustrate the concepts we have discussed.

## Related work

Fuzzy logic has been used in a wide variety of systems such as the automatic focus of digital cameras [10], control and optimization of industrial processes and systems [11], improving the efficiency of fuel-running engines [12], environmental improvement [13], expert systems [14], robotics [15], vehicles and autonomous driving [16], computer technology [17], Fuzzy databases [18], artificial intelligence, control systems for air conditioners [19], family appliances [3, 20], wireless sensor networks [3–6], and cellular automata [21–23].

Concerning formal verification of IoT systems, the authors in survey [24] present various works focused on verifying security properties [25–27]. Some other IoT works studied the settings of formal verification, including communication protocols [28], healthcare and environmental monitoring systems [29, 30]. Even when all these approaches focus on verifying data,

protocols, and security consistency, these proposals work over static variables. On the contrary, the proposal presented in this paper can modify the system's behavior by adding new rules on running time while the system verifies consistency. Furthermore, none of the above proposals interact with end users.

Input data in IoT systems is usually collected from heterogeneous sensor devices that need more interoperability since data values are based on proprietary formats. Similarly, IoT systems can accumulate poor-quality data since events such as offset data, missing data, wrong time stamps, and wrong attribute values can occur. Verifying the consistency of collected data has traditionally used machine learning and point-based calibration algorithms. For instance, authors in [31] proposed a data consistency method based on neural networks to reduce data errors by approximately 4%. However, this approach cannot interact in real-time with end users since it verifies consistency before the system starts.

Logical data inconsistencies have also been studied in the description logic (DLs) setting comprising a family of knowledge representation languages [32]. The balance between computational complexity and the expressiveness of DLs has allowed efficient reasoning tools to be constructed. These tools have enabled the application of DLs in several domains successfully [33]. Notably, the Web Ontology Language (OWL), a standard for Web Semantics technologies, is based on DLs [34]. Fuzzy extensions of DLs have also been developed [35]. These extensions have found application in human activity modeling for ambient intelligence systems [36], diabetes diagnosis systems [37], and database systems [38], to mention a few. Authors in [9] proposed a consistency data representation for IoT healthcare systems, transforming health data obtained from heterogeneous IoT devices into a semantic data model that supports logical reasoning using OWL. Even when the authors used a logic reasoner, they only focused on creating a unified static data model in which new rules cannot be introduced on running time. In [8], the authors proposed a reasoning framework to guarantee the consistency of the data stream produced by physical sensors in smart spaces. However, this framework does not interact with end users.

In summary, the proposed framework sets apart from previous works in the literature in two directions, mainly in the context of IoT applications. Firstly, it separates which tasks in the IoT system belong to an expert and which ones are the end user's responsibility, thus freeing end users from dealing with the most complex part of building and operating an IoT system. Most related works do not make this task distinction, providing little freedom to users wanting to implement their own IoT applications. However, the interaction of expert and user-related tasks will likely generate inconsistencies in the instructions introduced by end users and the data being collected and processed by sensors and actuators. Secondly, this framework verifies the dynamic properties of these task interactions and detects inconsistencies resulting from end-user instructions and wrong data that can be detected in real-time. This may allow end users to identify contradictory instructions so they can be modified to guarantee the IoT system's correctness. About this point, most related works dealing with consistency focus on verifying static properties defined for the design of the IoT system only, but they do not consider the dynamic aspect once the system is running.

## IoT systems

Fuzzy logic is a multi-valued logic whose statements can take truth values associated with the interval [0, 1] [39]. In deductive logic, inferences have the following structure:

$$\text{If } P_1, P_2, \ldots, P_n, \text{ then } R.$$

From the premises $P_1$, $P_2$, . . ., $P_n$, the conclusion $R$ is reached. Consider, for example, the following inferences in classical logic:

| $P_1$ : | All men are mortal | $P_1$ : | All cats like fish |
|---|---|---|---|
| $P_2$ : | Aristotle is a man | $P_2$ : | Mishi is a cat |
| $R$ : | Aristotle is mortal | $R$ : | Mishi likes fish. |

Notice that in these examples, the degrees of membership are Boolean: Aristotle is a man, and Mishi is a cat. However, in many contexts, the degrees of membership are not Boolean. For example, the route to the airport meets with much traffic, and the wind chill is hot. In fuzzy logic, the structure of the inferences is conserved concerning classical reasoning. The degrees of membership are those that are considered fuzzy. Consider the following example:

| $P_1$ : | The temperature is too high |
|---|---|
| $P_2$ : | The humidity is not low |
| $R$ : | Turn on the airconditioning at medium speed. |

Let us assume a set of sensors $\mathcal{S}$ and a set of actuators $\mathcal{A}$ in an IoT security system. For example, sensors might detect *movement*, *light*, and *sound*, while actuators make an action, such as set the *alarm*, *phone_call*, and *call_the_police*. Domains of sensors and actuators are considered to be fuzzy. This occurs because data obtained from sensors and actuators could be more accurate in practice due to many factors. A fuzzy domain $D$ is a finite set of closed intervals $d_1$, $d_2$, . . ., $d_n$, such that $d_i \subseteq [0, 1]$ and $\bigcup_{i=1}^{n} d_i = [0, 1]$. For instance, a domain for an alarm actuator can be defined by the intervals [0, 0.35] (low), [0.35, 0.7] (medium), and [0.7, 1] (high). Note that human language often uses adjectives instead of particular intervals.

To associate fuzzy domains with sensors and actuators, we consider fuzzy interpretation functions, that is, $f_z : \mathcal{S} \cup \mathcal{A} \mapsto D$, for a fuzzy domain $D$. Consider the example of an alarm actuator; if the alarm is set high, we formally write $f_z(alarm) \in [0.7, 1]$. It also can be written $f_z(alarm) \in$ high.

The following grammar defines a system expression:

$$\text{SystExp} := f(e) \in d \mid f(e) \notin d \mid \text{SystExp} \vee \text{SystExp} \mid \text{SystExp} \wedge \text{SystExp},$$

where $e \in \mathcal{S} \cup \mathcal{A}$. The system expression $f_z(alarm) \in$ high stands when the alarm is set to high. We may also write $f_z(alarm) \notin$ high to show that the alarm is not high. Other Boolean combinations of these atomic expressions may also form a system expression: $f_z(alarm) \in$ high $\wedge$ $f_z(lamp) \in$ on.

**Definition 1** (IoT System). Consider a set of fuzzy interpretation functions. We then define an IoT System as a finite set of rules of the following form:

$$\text{If SystExp then } f_z(a) \in d,$$

where SystExp are system expressions and $f_z$ are fuzzy interpretation functions of actuators $a$.

**Example.** Consider three sensors: *movement*, *light*, and *sound*. Domains for these sensors are composed of three intervals: few, some, and much. As for actuators, the system comprises *alarm* and *phone_call*. *Alarm* domain is defined by low, medium, and high. The domain for *phone_call* is Boolean; it can be set on or off. Now, the IoT system can be programmed to call the police if the sensor perceives much sound, light, and movement. The following expression

can express this behavior:

$$\text{If } f_z(movement) \in \text{much } \wedge f_z(light) \in \text{much } \wedge f_z(sound) \in \text{much}$$
$$\text{then } f_z(phone\_call) \in \text{on.} \tag{3}$$

If the system perceives some *movement* but low *sound* and some *light*, instead of calling the police, turn the alarm on to *medium*. This behavior can be written as follows:

$$\text{If } f_z(movement) \in \text{some } \wedge f_z(light) \in \text{few } \wedge f_z(sound) \in \text{few}$$
$$\text{then } f_z(alarm) \in \text{medium.} \tag{4}$$

Before providing semantics for IoT System expression, we must precisely define when a sensor or actuator is set to a particular interval domain. This is not immediate since intervals may intersect; for instance, this might be the case for medium and low intervals for a sound sensor. We then introduce the fuzzy membership function $m_f \colon D \mapsto [0, 1]$, provided a fuzzy domain $D$. Fuzzy membership functions may be defined according to the application context of the IoT System. Some standard Fuzzy membership functions are (a) trapezoidal, (b) triangle, (c) rectangular, (d) right-shoulder, and (e) left-shoulder as depicted in Fig 2.

The membership of a sensor or actuator to a particular domain is then defined as the maximum of the fuzzy membership functions. This is formalized by a Boolean structure, which is

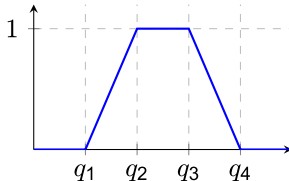

(a) Trapezoidal

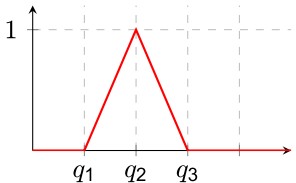

(b) Triangular

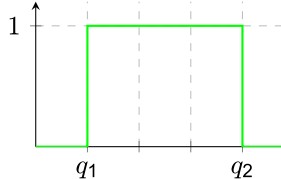

(c) Rectangular

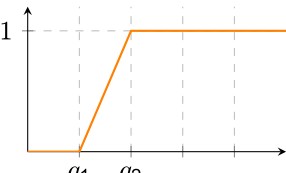

(d) Right-Shoulder

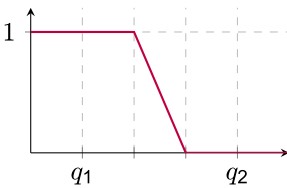

(e) Left-Shoulder

**Fig 2. Fuzzy membership functions.**

defined as a function from expressions $f(e) \in d$ to {0,1} as follows:

$$B(f(e) \in d) = 1, \text{ if and only if, } \max\{m_f(d) \mid d \in D\}.$$

Recall the example of the sound sensor intervals intersecting: if the sensed value falls within this intersection, our system employs a special structure to determine which interval is closer to the sensed value.

We are now ready to provide precise semantics of an IoT System. The interpretation of an IoT system rule concerning a Boolean structure $B$ is then defined as follows:

- $\llbracket f_z(e) \in d \rrbracket^B = 1$, if and only if, $B(f_z(e) \in d) = 1$;

- $\llbracket f_z(e) \notin d \rrbracket^B = 1$, if and only if, $B(f_z(e) \in d) = 0$;

- $\llbracket \text{SystExp}_1 \vee \text{SystExp}_2 \rrbracket^B = 1$, if and only if, $\llbracket \text{SystExp}_i \rrbracket^B = 1$ for some $i \in \{1, 2\}$;

- $\llbracket \text{SystExp}_1 \wedge \text{SystExp}_2 \rrbracket^B = 1$, if and only if, $\llbracket \text{SystExp}_i \rrbracket^B = 1$ for all $i \in \{1, 2\}$;

- $\llbracket \text{If SystExp then } f_z(a) \in d \rrbracket^B = 1$, if and only if, $\llbracket \text{SystExp} \rrbracket^B = 0$ or $\llbracket f_z(a) \in d \rrbracket^B = 1$.

The intuition of interpreting a system rule is that Boolean structures provide a particular context for sensors and actuators.

**Definition 2** (IoT System Consistency). We say an IoT System is consistent, if and only if, for all rules of the system $R$ and any Boolean structure $B$, have that $\llbracket R \rrbracket^B = 1$.

For instance, consider a system comprised of expressions 3 and 4. It is evident that this configuration is consistent, as both rules are interpreted as 1 under any Boolean structure (within the context of sensors and actuators). Additionally, if we include the following rule:

$$\begin{aligned} & \text{If } f_z(movement) \notin \text{much } \wedge f_z(light) \in \text{few } \wedge f_z(sound) \in \text{few} \\ & \text{then } f_z(alarm) \in \text{low.} \end{aligned} \tag{5}$$

Then the system is inconsistent, as there is a Boolean structure considering sensors detecting some movement, low light, and sound, and setting the alarm to medium, such that the interpretation of rule 5 is 0. Note that there is another structure considering the alarm set to low, which causes rule 5 to hold. However, under this structure, rule 4 does not.

## Fuzzy description logic verification

In this Section, we describe a fuzzy description logic. Description logics form a family of Knowledge Representation languages, vastly and successfully known among several other domains in the Semantic Web community. The description logic language described in this work allows us to model IoT expert systems in the form of Knowledge Bases (KB). The fuzzy part of language allows us to model ambiguous notions such as "much movement". We will first describe the syntax and semantics of the logic. Then, the notion of KB consistency is introduced. We next show that the consistency of IoT systems can be tested in terms of KB consistency.

Zadeh [39] proposed fuzzy set theory and logic to manage fuzzy and ambiguous knowledge. In classical set theory, either one of the elements belongs to the set, or it does not. In fuzzy set theory, the elements belong to a certain degree. Let $X$ be a set of elements called the reference set. A fuzzy subset $A$ of $X$ is defined by a membership function $\mu_A(x)$ (or simply $A(x)$), which assigns any $x \in X$ to a value in the real interval between 0 and 1. In the classical case, 0 is no membership, and 1 is full membership. It should be noted that the fuzzy interpretation functions defined in the previous section and the membership functions exhibit similar behavior. The distinction lies in the domain that each considers. To avoid confusion, the former

functions pertain to the IoT system, while the latter pertains to fuzzy description logic. A value between 0 and 1 indicates how $x$ is considered an element of $X$. The crisp set operation is extended to fuzzy sets. The intersection, union, complement operation, and implication set are interpreted as a t-norm $\otimes$, a t-conorm $\oplus$, a negation $\ominus$, and an implication $\Rightarrow$. Let $\alpha, \beta \in [0, 1]$, we define the fuzzy operators' negation, t-norm, t-conorm, and implication; moreover, the fuzzy implications are used to describe the Zadeh logic: $\alpha \otimes \beta = min\{\alpha, \beta\}$, $\alpha \oplus \beta = max\{\alpha, \beta\}$, $\ominus \alpha = 1 - \alpha$, and $\alpha \Rightarrow \beta = max\{1 - \alpha, \beta\}$. It is possible to define more operators to define Łukasiewicz logic, Kleene-Dienes logic, and Classical logic (for more details see [35]). The Mamdani model is a particular and usual case of reasoning in the literature. This is a fuzzy **If-Then** system that includes the basic rule, that is, a set of rules of the form:

$$\textbf{If } X_1 \text{ is } A_1 \text{ and } X_n \text{ is } A_n \textbf{ Then } Y \text{ is } B, \tag{6}$$

where for each $i = 1, \ldots, n$, $A_i$, and $B$ are linguistic values defined by language-wide fuzzy sets. $X_i$ and $Y$, respectively. The inference rule (Generalized Modus Ponens). Note that the Mamdani model is a particular case of Definition 1.

We introduce the fuzzy description logic behind *fuzzyDL* inference engine. *fuzzyDL* is defined on a discrete set $[0, 1]_D = \{0, \frac{1}{n}, \ldots, \frac{n-1}{n}, 1\}$ for $n \in \mathbb{N}$ such that there is a machine representable $0 < \epsilon < \frac{1}{n}$.

**Definition 3**. Fuzzy Description Logic's main elements are concepts, denoting unary predicates, and roles, denoting binary predicates. Finally, connectives allow the construction of complex concepts. The syntax of *fuzzyDL* is as follows:

$$
\begin{aligned}
C, D &:= & \top \mid & \bot \mid & \neg C \mid & C \sqcup_\circ D \mid & C \sqcap_\circ D \mid \\
& C \rightarrow_\circ D \mid & \forall R.C \mid & & \exists R.C \mid & \forall T.d \mid & \exists T.d \mid \\
d &:= & L(a, b) \mid & R(a, b) \mid & crisp(a, b) \mid & & trapezoidal(a, b, c, d) \\
m &:= & linear(a) \mid & & triangular(a, b, c),
\end{aligned}
$$

where C, D denote concepts, $R$ denotes abstract roles names, $T$ denotes concrete roles names, $d$ denotes membership functions, $m$ denotes modifiers, and $\circ = \{L, G\}$.

**Example.** Consider again the IoT system rule 4

$$\text{If } f_z(movement) \in \text{some } \wedge f_z(light) \in \text{few } \wedge f_z(sound) \in \text{few}$$
$$\text{then } f_z(alarm) \in \text{medium}.$$

In terms of a fuzzy description logic expression, this rule is written as follows:

$$(\exists move.SomeMove \sqcap \exists light.LowLight \sqcap \exists sound.FewSound) \rightarrow \exists alarm.MediumAlarm$$

**Definition 4**. A fuzzy knowledge base (KB) consists of: $\mathcal{A}$ is a fuzzy ABox, $\mathcal{T}$ is a fuzzy TBox, and $\mathcal{R}$ is a fuzzy RBox denoted by $K = \langle \mathcal{A}, \mathcal{T}, \mathcal{R} \rangle$, where:

- A fuzzy ABox $\mathcal{A}$ consists of a finite set of fuzzy concepts and fuzzy role assertion axioms of the form $\langle x : C, \alpha \rangle$ and $\langle (x, y):R, \alpha \rangle$, where $\alpha \in (0, 1]_D$.

- A fuzzy TBox $\mathcal{T}$ is a finite set of fuzzy General Concept Inclusion axioms (GCIs) $\langle C \sqsubseteq D, \alpha \rangle$, where $C, D$ are concepts and $\alpha \in (0, 1]_D$. Informally, it states that all instances of concept $C$ are instances of concept $D$ to degree $\alpha$; that is, the subsumption degree between $C$ and $D$ is at least $\alpha$. We write $C = D$ as a shorthand of the two axioms $\langle C \sqsubseteq D, 1 \rangle$ and $\langle D \sqsubseteq C, 1 \rangle$.

- A fuzzy RBox $\mathcal{R}$ is a finite set of role axioms of the form: (*funR*) a role $R$ is functional, (*transR*) a role $R$ is transitive, $R_1 \sqsubseteq R_2$ role $R_2$ subsumes role $R_1$.

In the setting of an IoT system, the information commonly provided by an expert, such as fuzzy interval for sensor and actuator (much movement, low light, etc.) is described as an ABox. Whereas the information corresponding to rules (instructions) provided by the end user are codified in terms of TBox expressions.

We now introduce the semantic notions of the fuzzy description logic.

A fuzzy data type $\mathbf{D} = \langle \Delta_{\mathbf{D}}, \cdot_{\mathbf{D}} \rangle$ is such that $\cdot_{\mathbf{D}}$ assigns to every n-ary fuzzy relation over $\Delta_{\mathbf{D}}$. For instance, the predicate $\geq_0$ may be a crisp unary predicate over $\mathbb{R}$, denoting the set of reals smaller or equal to 0.

**Definition 5.** A fuzzy interpretation $\mathcal{I} = (\Delta^{\mathcal{I}}, \cdot^{\mathcal{I}})$ relative to a fuzzy data type theory $\mathbf{D} = (\Delta_{\mathbf{D}}, \cdot)_{\mathbf{D}}$ consists of a nonempty set $\Delta^{\mathcal{I}}$ disjoint from $\Delta_{\mathbf{D}}$ and of a fuzzy interpretation function $\cdot^{\mathcal{I}}$ that coincides with $\cdot^{\mathcal{I}}$ on every data value, data type, and fuzzy data type predicate, and it assigns:

- To each abstract concept $C$ a function $C^{\mathcal{I}} : \Delta^{\mathcal{I}} \to [0, 1]_D$.

- To each abstract role $R$ a function $R^{\mathcal{I}} : \Delta^{\mathcal{I}} \times \Delta^{\mathcal{I}} \to [0, 1]_D$.

- To each abstract feature $r$ a partial function $r^{\mathcal{I}} : \Delta^{\mathcal{I}} \times \Delta^{\mathcal{I}} \to [0, 1]_D$ such that for all $x \in \Delta^{\mathcal{I}}$ there is a unique $y \in \Delta^{\mathcal{I}}$ on which $r^{\mathcal{I}}(x, y)$ is defined.

- To each concrete role $T$ a function $R^{\mathcal{I}} : \Delta^{\mathcal{I}} \times \Delta_D \to [0, 1]_D$

- To each concrete feature $t$ a partial function $t^{\mathcal{I}} : \Delta^{\mathcal{I}} \times \Delta_D \to [0, 1]_D$ such that for all $x \in \Delta^{\mathcal{I}}$ there is a unique $v \in \Delta_D$ on which $t^{\mathcal{I}}(x, v)$ is defined.

- To each modifier $m$ the modifier function $f_m : [0, 1]_D \to [0, 1]_D$.

- To each abstract individual $x$ an element in $\Delta^{\mathcal{I}}$.

- To each concrete individual $v$ an element in $\Delta_D$.

**Definition 6.** The mapping $\cdot^{\mathcal{I}}$ is extended to roles and complex concepts as follows:

- $\perp^{\mathcal{I}}(x) = 0$, $\top^{\mathcal{I}}(x) = 1$

- $(\neg C)^{\mathcal{I}}(x) = \ominus C^{\mathcal{I}}(x)$

- $(C \sqcap_\circ D)^{\mathcal{I}}(x) = C^{\mathcal{I}}(x) \otimes_\circ D^{\mathcal{I}}(x)$ where $\circ = \{C, G, L\}$

- $(C \sqcup_\circ D)^{\mathcal{I}}(x) = C^{\mathcal{I}}(x) \oplus_\circ D^{\mathcal{I}}(x)$ where $\circ = \{C, G, L\}$

- $(C \to_* D)^{\mathcal{I}}(x) = C^{\mathcal{I}}(x) \Rightarrow_* D^{\mathcal{I}}(x)$ where $* = \{C, G, KD, L\}$

- $(\exists R.C)^{\mathcal{I}}(x) = \sup_{y \in \Delta^{\mathcal{I}}} R^{\mathcal{I}}(x, y) \otimes C^{\mathcal{I}}(y)$.

**Definition 7.** The satisfaction of a fuzzy axiom $E$ by a fuzzy interpretation $\mathcal{I}$, denoted $\mathcal{I} \models E$, is defined as:

- $\mathcal{I} \models \langle \tau \geq \alpha \rangle$ if and only if $\tau^{\mathcal{I}} \geq \alpha$.

- $\mathcal{I} \models \langle \textit{trans } R \rangle$ if and only if $\forall x, y \in \Delta^{\mathcal{I}}, R^{\mathcal{I}}(x, y) \geq \sup_{z \in \Delta^{\mathcal{I}}} R^{\mathcal{I}}(x, z) \otimes R^{\mathcal{I}}(z, y)$.

- $\mathcal{I} \models R_1 \sqsubseteq R_2$ if and only if $\forall x, y \in \Delta^{\mathcal{I}}, R_1^{\mathcal{I}}(x, y) \leq R_2^{\mathcal{I}}(x, y)$.

- $\mathcal{I} \models (\textit{inv } R_1 \, R_2)$ if and only if $\forall x, y \in \Delta^{\mathcal{I}}, R_1^{\mathcal{I}}(x, y) = R_2^{\mathcal{I}}(y, x)$.

The concept $C$ is satisfiable if and only if there is an interpretation $\mathcal{I}$ and an individual $x \in \Delta^{\mathcal{I}}$ such that $C^{\mathcal{I}}(x) > 0$.

Let $F$ be a set of axioms and $E$ be a fuzzy axiom. We will say that $\mathcal{I}$ satisfies $F$ if and only if $\mathcal{I}$ satisfies each axiom in $F$. $\mathcal{I}$ is a model of $E$, if and only if, $\mathcal{I} \models E$ and $\mathcal{I}$ is a model of $F$, if and only if, $\mathcal{I} \models F$. $\mathcal{I}$ is a model of KB, if and only if $\mathcal{I}$ is a model of each component $\mathcal{A}$, $\mathcal{T}$ and $\mathcal{R}$, denoted by $\mathcal{I} \models K$. An axiom $E$ is a logical consequence of a knowledge base $K$, if and only if every model of $K$ satisfies $E$, denoted by $K \models E$. Finally, a fuzzy knowledge base $K$ is consistent iff a model of $K$ satisfies each axiom. This notion of consistency for fuzzy description logic knowledge base is equivalent to the notion of consistency of IoT systems. In order to formally prove this equivalence, we first define a translation function $(\cdot)^*$ from IoT systems to knowledge bases.

**Definition 8**. Let $(\cdot)^* :$ SystExp $\rightarrow$ *FDL* be a star-interpretation from SystExp(system expressions) to FDL (fuzzy description logic) defined as:

- $(f_z(a) \in d)^* = \exists A.D$

- $(f_z(a) \notin d)^* = \neg(\exists A.D)$

- $(\text{SystExp}_1 \wedge \text{SystExp}_2)^* = (\text{SystExp}_1)^* \sqcap (\text{SystExp}_2)^*$

- $(\text{SystExp}_1 \vee \text{SystExp}_2)^* = (\text{SystExp}_1)^* \sqcup (\text{SystExp}_2)^*$

- $(\text{If SystExp then } f_z(a) \in d)^* = (\text{SystExp})^* \rightarrow (\exists A.D)$

    **Example.** Note that IoT system rules 4 and 5 are translated as follows:

    $(\exists \text{move.SomeMove} \sqcap \exists \text{light.LowLight} \sqcap \exists \text{sound.FewSound}) \rightarrow \exists \text{alarm.MediumAlarm}$

    $(\exists \text{move.} \neg \text{MuchMove} \sqcap \exists \text{light.LowLight} \sqcap \exists \text{sound.FewSound}) \rightarrow \exists \text{alarm.LowAlarm}$

This Knowledge Base is inconsistent since no model satisfies both axioms.

We now state the main formal result of the article: IoT system consistency can be tested in terms of Knowledge Base consistency. In practice, consistency is tested before the execution of the IoT system. The following Theorem provides a mathematical guarantee that the system is free of inconsistencies under any real-time scenario (any sensor inputs).

**Theorem 1**. *An IoT System $R_1, R_2, \ldots, R_n$ is consistent, if and only if there is a Knowledge Base $K$, such that $K \models R_1^* \sqcap R_2^* \sqcap \cdots \sqcap R_n^*$.*

*Proof.*

**"Only if"** part. Suppose that the IoT system $R_1$ is consistent. Note that $R_1 = \bigwedge_{i=1}^{n}(\text{SystExp}_i)$, then by applying the star-interpretation, we obtain

$$R_1^* = \left(\bigwedge_{i=1}^{n}(\text{SystExp}_i)\right)^* = \sqcap_{i=1}^{n}(\text{SystExp}_i)^*$$

**Claim:** There is an interpretation $\mathcal{I}$ and $x \in \Delta^{\mathcal{I}}$ such that $(R_1^*)^{\mathcal{I}}(x) > 0$. Indeed, otherwise for each interpretation $\mathcal{I}$ and $x \in \Delta^{\mathcal{I}}$ such that $(R_1^*)^{\mathcal{I}}(x) \leq 0$. Without loss of generality suppose that for each $j \in [1, n-1]$, $(\text{SystExp}_j^*)^{\mathcal{I}}(x) \neq 0$ and $(\text{SystExp}_n^*)^{\mathcal{I}}(x) = 0$, consequently $\text{SystExp}_n = f_z(a) \notin d$. Then, given a Boolean structure $B$, we have that $[\![R_1]\!]^B = [\![\bigwedge_{j=1}^{n}(\text{SystExp}_j) \wedge \text{SystExp}_n]\!]^B = 0$, thus the IoT system is not consistent, which is a contradiction. We conclude that there is a knowledge base $K$ that satisfies $K \models R_1^*$.

Suppose that the IoT system $R_1, R_2$ is consistent, where $R_1$ is as in the previous step and $R_2 = \text{If } R_1 \text{ then } A$.

**Claim:** $K \models A^*$. Indeed, otherwise for each interpretation $\mathcal{I}$ and $x \in \Delta^{\mathcal{I}}$ such that $(A^*)^{\mathcal{I}}(x) \leq 0$ consequently, $A = (f_z(a) \notin d)$. Given a Boolean structure B, we have that

$[\![R_2]\!]^B = [\![$If $R_1$ then $A]\!]^B = 0$, which is a contradiction. Therefore, the knowledge base $K$ that satisfies $K \models R_1{}^* \sqcap R_2{}^*$. Following the previous construction, it is concluded that if $R_1, \ldots, R_n$, $R_{n+1}$ is consistent, then $K \models R_1{}^* \sqcap \cdots \sqcap R_n{}^* \sqcap R_{n+1}{}^*$.

**"If"** part. Suppose there is a knowledge base $K$ that satisfies $K \models R_1{}^* \sqcap R_2{}^* \sqcap \cdots \sqcap R_n{}^*$ and that the IoT system $R_1, R_2, \ldots, R_n$ is not consistent. Since $K \models R_1{}^* \sqcap R_2{}^* \sqcap \cdots \sqcap R_n{}^*$ then every model $\mathcal{I}$ and each $x \in \Delta^{\mathcal{I}}$ verify that: $(R_1{}^* \sqcap R_2{}^* \sqcap \cdots \sqcap R_n{}^*)^{\mathcal{I}}(x) > 0$.

$$(R_1{}^* \sqcap R_2{}^* \sqcap \cdots \sqcap R_n{}^*)^{\mathcal{I}}(x) = (R_1{}^*)^{\mathcal{I}}(x) \otimes (R_2{}^*)^{\mathcal{I}}(x) \otimes \cdots \otimes (R_n{}^*)^{\mathcal{I}}(x)$$
$$= \min\{(R_i{}^*)^{\mathcal{I}}(x) \mid i \in [1, n]\} = 1$$

So for each $i \in [1, n]$, $(R_i{}^*)^{\mathcal{I}}(x) = 1$ and when considering the inverse interpretation, we have that for each $(R_i{}^*)^{-1} = R_i$ such that $[\![R_i]\!]^B = 1$, where $B$ is any Boolean structure. Therefore, the IoT system $R_1, R_2, \ldots, R_n$ is consistent, which is a contradiction.

Once the IoT system consistency is verified in terms of Knowledge Base consistency, the system can be executed with the guarantee that no errors can be computed, no matter the inputs from sensors. Values for actuators can be computed from the instructions in the Knowledge Base by means of a defuzzification process. Defuzzification is the output value for the membership function $m$ on the values of the variables in $x$ using the specified defuzzification method. Some examples of defuzzification methods can be seen in the following definition.

**Definition 9**. Let $B$ be a fuzzy set to be defuzzified, and let $x$ be an arbitrary element of the universe. Then for all $x$:

- $x_{LOM}$ is the largest of maxima (**LOM**), if and only if, $\mu_\beta(x_{LOM}) \geq \mu_\beta(x)$, and if $\mu_\beta(x_{LOM}) = \mu_\beta(x)$ then $x_{LOM} > x$.

- $x_{SOM}$ is the smallest of maxima (**SOM**), if and only if, $\mu_\beta(x_{SOM}) \geq \mu_\beta(x)$, and if $\mu_\beta(x_{SOM}) = \mu_\beta(x)$ then $x_{SOM} < x$.

- $x_{MOM}$ is the middle of maxima (**MOM**), if and only if, $x_{MOM} = \dfrac{(x_{LOM} + x_{SOM})}{2}$.

Consider for instance, in the following axiom: ($\exists$move.SomeMove$\sqcap\exists$light.LowLight$\sqcap\exists$sound.FewSound) $\rightarrow \exists$alarm.MediumAlarm. If input sensors for movement, light, and sound correspond to some, low, and few, respectively, then the alarm should be set to medium. The numerical value corresponding to medium is computed by the defuzzification process.

## Fuzzy control for an IoT security system

Smart-home systems are challenging when implementing security systems since wireless sensor devices used in IoT can be heterogeneous and use various communication protocols having different coverage areas when detecting intruders or risk situations. Moreover, even when intelligent IoT devices can monitor their sensors to notify users about potential issues or risks in smart homes, most applications operate without knowledge-based consistency, provoking the system to take wrong actions or presenting failures since more than one rule can contradict each other. For instance, suppose a smart-security home application with sensors that measure light, movement, and sound, and a user is looking for security against an intruder. This system, guided by (a) blueprint, strategically places sensors in the (b) hall, (c) dining room, (d) library, and (e) living room (see small black sensors placed on the walls in Fig 3). The goal is to create a secure and comfortable (f) Home (refer to its configuration). A *fuzzyDL* reasoner is a system

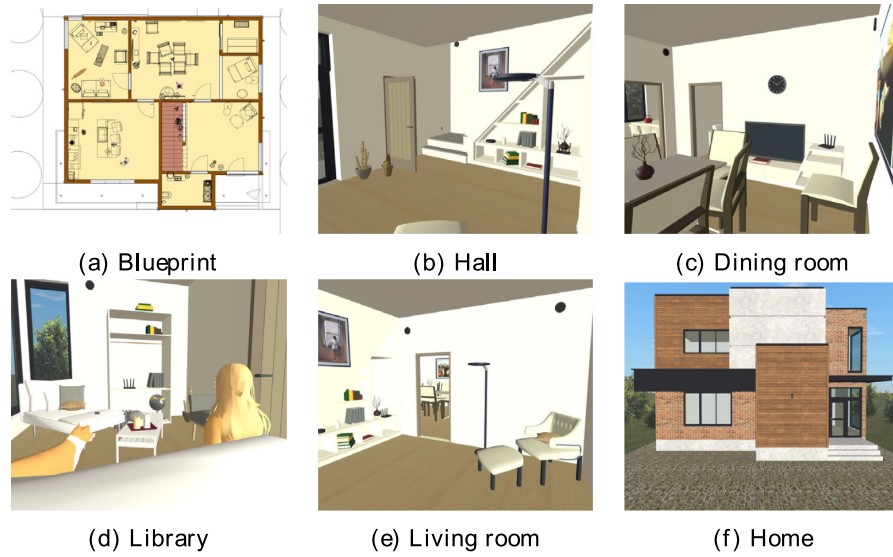

**Fig 3. Smart home (black dots are wall sensors).**

in which a user establishes the rules to take actions, for example, calling the police, sending a notification message, or turning on the alarm.

Let us assume that an expert in security systems has pre-programmed the IoT System with certain restrictions on the RBox, TBox, and Datatypes (Step 1). Since the user can define the TBox with basic rules according to their preferences (Step 2), the objective of the *fuzzyDL* tool is to verify that the TBox rules are consistent. At the same time, the actuators operate according to the rules established by the user (Step 3). This process involves checking the program syntax and conducting experiments (Program syntax, Experiments) to ensure the smooth running of the system.

## Step 1. Contextual information

An expert plays a crucial role by providing specifications regarding the specific context in which IoT systems are implemented. For instance, in the case of a smart home security system, this step meticulously defines what constitutes "high movement" in terms of the numerical data obtained from sensors. Notably, certain information of this nature can be influenced by user preferences, enabling users to modify pre-programmed contextual details. For instance, the interpretation of "hot temperature" might vary between Nordic users and their tropical counterparts. Additionally, finer details, such as the characteristics of fuzzy membership functions (triangular, left shoulder, etc.), are also meticulously delineated in this phase. These precise definitions correspond to the ABox in the corresponding Fuzzy Description Logic Knowledge Base.

The heart of the system lies in its user-friendly interface, designed to gather contextual information effortlessly. This interface adeptly processes natural language instructions through voice or text. To enrich the user experience further, the interface portrays information about sensor types, actuators, and domain values.

Simultaneously, the role of the expert encompasses determining the behavior of sensors and actuators alongside furnishing initial programming for the system. An inherent assumption in this context is the consistency of rules established by the expert. As for system sensors,

this illustrative example features three input systems meticulously defined by the expert: light, movement, and sound. These inputs collaboratively contribute to the computation of an output value, which subsequently triggers alert mechanisms.

**System sensors.** The system, in this instance, encompasses three input systems meticulously outlined by the expert: light, movement, and sound. These inputs harmoniously collaborate to calculate an output value that triggers alert mechanisms.

- Light is associated with three labels: low, medium, and high. For example, LowLight, the label of low light, can be defined as a triangular membership function ($q_1,q_2,q_3$).

- Movement has five labels: low, middle, high, and very high. For example, LowMovement, the label representing low movement, can be defined as a triangular membership function ($q_1,q_2,q_3$).

- sound has five labels: low, middle, high, and very high. For example, LowSound, the label of low sound, can be defined as a triangular membership function ($q_1,q_2,q_3$).

**Actuator system.** Four actuators were designated with different colors: Green, Yellow, Orange, and Red. For instance, each color corresponds to a specific action, making it a clearer and more precise description of the color-to-action assignment.

- Green: everything is in order.

- Yellow: sending an alert to a cell phone.

- Orange: sending an alert to the police.

- Red: taking further action.

## Step 2. System rules

The system's rules may be pre-programmed, but ideally, non-expert users are expected to define particular rules for the system. Considering the smart home security system, examples are: if movement, light, and sound are low, then do nothing; or if movement, light, and sound are high, then call the police. These instructions correspond to the TBox of the corresponding Fuzzy Description Logic Knowledge Base. A user-friendly interface is also considered for this step: voice or text instructions directly from the users, and information about the system (sensors, actuators, etc.) are depicted to help users define these instructions. At this step, a logic reasoner analyzes the instructions to detect inconsistencies: if the user provides an instruction contradicting an already loaded instruction, the interface provides a warning. In other words, the logic reasoner guarantees that the rules the user introduces are consistent under any system setting. On the other hand, the user's role is to indicate the rules (the TBox) that satisfy an ideal security system. Furthermore, the consideration of system rules. The number of permutations determined by the labels of each sensor corresponds to the following arithmetic operation $3 \times 4 \times 4$ (for the previous example). Therefore, the system rule-set is 48 rules. Let us assume that the user determined the following four rules:

R1. IF light IS low AND movement IS low AND sound IS low, THEN code AlertGreen.

R2. IF light IS low AND movement IS low AND sound IS middle, THEN code AlertYellow.

R3. IF light IS low AND movement IS low AND sound IS high, THEN code AlertOrange.

R4. IF light IS low AND movement IS low AND sound IS very-high, THEN code AlertRed.

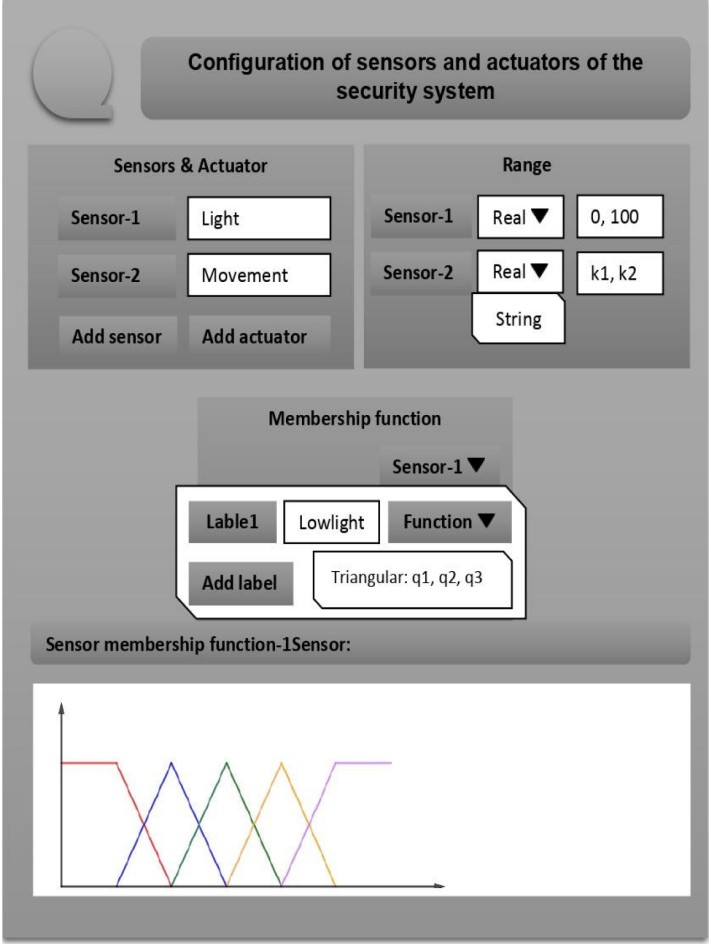

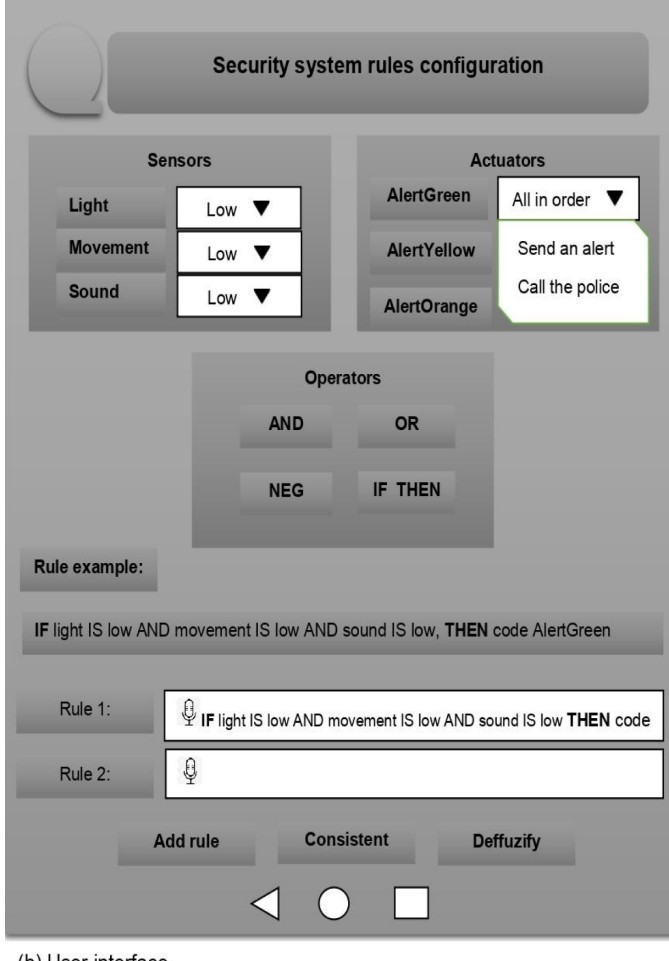

**Fig 4. Interfaces.**

The interfaces for step 1 (Contextual information) and step 2 (System rule) can be seen in Fig 4, with (a) representing the expert interface and (b) representing the user interface.

## Step 3. Running the system

Once the contextual information and consistent system rules are defined, then the system is executed. The logical reasoner fuzzyDL consists of three steps: (a) Fuzzification: The numeric inputs (light, movement, sound, light change, movement change, sound change) will be translated into linguistic values (fuzzy sets). (b) Fuzzy inference: Fuzzy rules will be applied to determine how much adjustment is necessary for the security system. (c) Defuzzification: This converts fuzzy outputs obtained from the inference step (Step 2) into a crisp or numerical value. In other words, it transforms the fuzzy sets and their degrees of membership into a single numerical result that represents the system's output or action. The fuzzy results obtained from the inference step are translated into a numerical action to adjust the security system. One common approach for defuzzification is the **MOM** method. Finally, when the sensors' current light, motion, and sound are input into the system, the fuzzy rules will be applied, and

the defuzzification will provide a numerical value indicating the action the actuator will follow for the optimum security system.

## Program syntax

The internal architecture of the code of our security system programmed in *fuzzyDL* reasoner is the following:

- System sensors. For each system variable (sensor), we define some specific characteristics that represent it. We also specify its range as a closed subset of the real ones [$k_1$,$k_2$], for example:
  (functional sensor-1)
  (functional sensor-2)
  (functional code)
  (range sensor-1 *real* $k_1$ $k_2$)
  (range sensor-2 *real* $k_3$ $k_4$)
  (range code *real* $k_5$ $k_6$)
  We define the linguistic labels to describe the value of these variables (using the triangular membership function, see Fig 5), such as:
  **For sensor-1**:
  (define-fuzzy-concept label1 triangular($k_1$ $k_2$ $q_1$ $q_2$ $q_3$))
  (define-fuzzy-concept label2 triangular($k_1$ $k_2$ $q_4$ $q_5$ $q_6$))
  **For sensor-2**:
  (define-fuzzy-concept label1 triangular($k_3$ $k_4$ $q_1$ $q_2$ $q_3$))
  (define-fuzzy-concept label2 triangular($k_3$ $k_4$ $q_4$ $q_5$ $q_6$))

- Actuator system. We define linguistic labels to describe the value of actuators.
  **For actuator-1**:
  (define-fuzzy-concept label1 triangular($k_1$ $k_2$ $q_1$ $q_2$ $q_3$))
  For example, AlertGreen, the label representing that the actuator green is in order, can be defined as triangular ($q_1$,$q_2$,$q_3$).

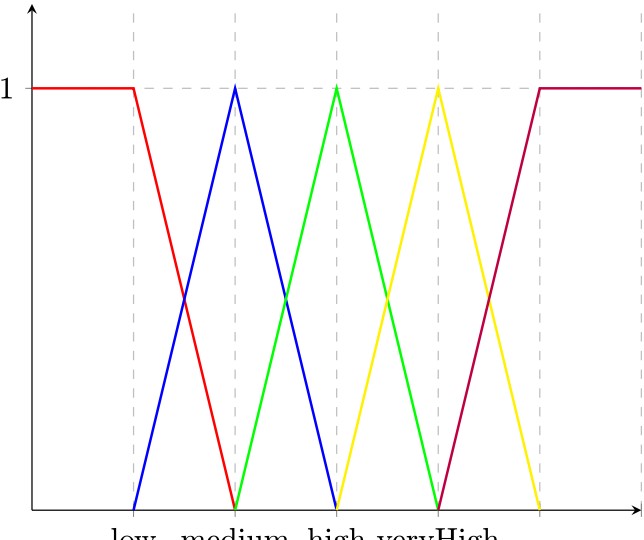

**Fig 5. Partitioning a domain using fuzzy membership functions.**

We represent the system input as fuzzy statements involving a single digest.
(instance individual (= sensor-1 $q'$) $\alpha$), where $q' \in [k_1, k_2]$
(instance individual (= sensor-2 $q''$) $\beta$) where $q'' \in [k_1, k_2]$

- System rules. The system defines each concept from the rules given by the user, for example:
R1. (define-concept Rule1(g-and(some sensor-1 label1)(some sensor-2 label2)))
R2. (define-concept Rule2(g-and(some sensor-1 label1)(some sensor-2 label2)))
RuleMamd. (define-concept Mamd (g-or Rule1 Rule2))

- Defuzzification. The output of the system is resolved through the use of queries.
(defuzzify-mom? Mamd individual sensor-3)
(sat RuleMamd)

The complete internal algorithm of the security system is depicted in Fig 6.

## Experiments

We conducted a comprehensive analysis of defuzzification and consistency within our security system. In **Experiment one** (see Table 1), we carried out the following tasks:

1. Introducing three distinct inputs for light, movement, and sound within the range defined by the expert. We then evaluated the defuzzification (**MOM**) and consistency (**sat**) queries. The outcomes aligned with our expectations.

2. For the fourth set of inputs for light, movement, and sound, we deliberately exceeded the predefined range, resulting in an inconsistency flagged by the **sat** query.

3. Lastly, we introduced an additional rule aimed at creating a contradiction within the system, which also resulted in inconsistency as indicated by the **sat** query.

4. Additionally, it is worth noting that the execution times in these experiments exhibited variability. This variability can be attributed to the specific configurations tested, particularly the introduction of contradictory rules that could either halt or significantly extend the program's execution. These time discrepancies underscore the influence of experimental factors on the system's performance.

Finally, in **Experiment two** (see Table 2), we programmed the three rules outlined in the IoT system section. In the program syntax, R2, R4, and R6 correspond to Eqs 3, 4 and 5, respectively. Our findings are as follows:

1. The **(sat)** query for rules R2 and R4 yielded a consistent outcome.

2. However, the **(sat)** query involving rules R2, R4, and R6 yielded inconsistency, primarily due to a contradiction between R4 and R6.

3. Additionally, the execution times in **Experiment two** exhibited variation. Notably, The execution of rule R7 showed consistency within 3 seconds. However, when involving rules R8, the program's execution time was extended to 10 minutes, reflecting inconsistency primarily caused by the interaction between rules R4 and R6.

The initial and final times were programmed on a computer with the following specifications: Computer Model: [Compaq nc6400]. Processor: [Genuine Intel(R) CPU T2300 1.66Ghz]. RAM: [2.50 GB]. Operating System: [Windows 7 Home Premium].

```
1  #Fuzzy logic
2  (define-fuzzy-logic zadeh)
3  #RBox axioms
4  (functional light)
5  (range light *real* 0 100)
6  (functional movement)
7  (range move *real* 0 100)
8  (functional sound)
9  (range sound *real* 0 100)
10 (functional code)
11 (range code *real* 0 9)
12 #Datatypes
13 (define-fuzzy-concept LowLight triangular(0,100,10,20,30))
14 (define-fuzzy-concept MidLight triangular(0,100,25,35,45))
15 (define-fuzzy-concept HighLight triangular(0,100,40,50,60))
16
17 (define-fuzzy-concept LowMovement triangular(0,100,5,15,25))
18 (define-fuzzy-concept MidMovement triangular(0,100,17,27,37))
19 (define-fuzzy-concept HighMovement triangular(0,100,32,42,52))
20 (define-fuzzy-concept VeryHighMovement triangular
       (0,100,47,57,67))
21
22 (define-fuzzy-concept LowSound triangular(0,100,40,50,60))
23 (define-fuzzy-concept MidSound triangular(0,100,55,65,75))
24 (define-fuzzy-concept HighSound triangular(0,100,70,80,90))
25 (define-fuzzy-concept VerySound triangular(0,100,85,95,100))
26
27 (define-fuzzy-concept AlertGreen triangular(0,9,1,2,3))
28 (define-fuzzy-concept AlertYellow triangular(0,9,2,3,4))
29 (define-fuzzy-concept AlertOrange triangular(0,9,4,5,6))
30 (define-fuzzy-concept AlertRed triangular(0,9,6,7,8))
31 #TBox axioms
32 (define-concept Rule1 (g-and (some light LowLight)(some move
       LowMovement)   (some sound LowSound)(some code AlertGreen)))
33 (define-concept Rule2 (g-and (some light MidLight)(some move
       MidMovement)   (some sound MidSound)(some code AlertYellow))
       )
34 (define-concept Rule3 (g-and (some light HighLight)(some move
       HighMovement)  (some sound HighSound)(some code AlertOrange))
       )
35 (define-concept RuleMamd (g-or Rule1 Rule2 Rule3))
36 #ABox axioms
37 (instance run1 (= light 20))
38 (instance run1 (= move 15))
39 (instance run1 (= sound 50))
40 #Query
41 (defuzzify-mom? RuleMamd run1 code)
42 (sat RuleMamd)
```

**Fig 6. The internal architecture of a security system code programmed in *fuzzyDL*.**

## Conclusion

The conclusions drawn from the analysis presented in the preceding sections fall into two main categories: Fuzzy Logic and IoT systems. First, we derived a theorem that establishes a relationship between the consistency of Fuzzy Logic and IoT systems, specifically in the context of the FuzzyDL reasoner. Second, we developed an algorithm for a security system that

**Table 1. Defuzzification and consistency: Experiment one.**

| Input | | | | Output | |
|---|---|---|---|---|---|
| Light | Movement | Sound | Query | Output-System | Description |
| 20 | 15 | 50 | MOM | Middle of the maxima defuzzification of feature code for instance run1 = 2.0 | If the system values are given as the peaks of the triangular function that correspond to the label low, then the defuzzification's MOM is equal to 2. I.T. 13:58:48:79 F.T. 13:58:50:91 **Means:** everything is in order. |
| | | | Sat | sat subsumes RuleMamd? $< = 1.0$ | The rules are consistent. |
| 30 | 22 | 60 | MOM | Middle of the maxima defuzzification of feature code for instance run1 = 3.0 | If the system values are given as the peaks of the triangular function that correspond to the label medium, then the MOM is equal to 3. I.T. 14:05:02:79 F.T. 14:05:04:55 **Means:** send an alert to the user's cell phone. |
| | | | Sat | sat subsumes RuleMamd? $< = 1.0$ | The rules are consistent. |
| 59 | 51 | 88 | MOM | Middle of the maxima defuzzification of feature code for instance run1 = 5.0 | If the input values are to the right of the peak of triangular function, then the MOM is equal to 5.0 14:10:02:80 F.T. 14:10:04:75 **Means:** send an alert to the police. |
| | | | Sat | sat subsumes RuleMamd? $< = 1.0$ | The rules are consistent. |
| 101 | 101 | 101 | Sat | KnowledgeBase inconsistent: Answer is 1.0. | The input values exceed the range, then the system is inconsistent. 14:13:29:61 F.T. 14:13:30:36 |
| Now if we add a new (contradictory) rule, namely, (define-concept Rule 4 (g-and (some light HighLight)(some move HighMovement) (not (some sound HighSound))(some code AlertOrange))) (define-concept RuleMamd (g-or Rule1 Rule2 Rule3 Rule4)) | | | | | sat subsumes RuleMamd? $< = 0.0$. 14:15:59:70 F.T. 14:16:00:45 **Means:** The rules are inconsistent. |

employs fuzzy logic as its primary language and identifies inconsistencies between rules. This system can find practical applications in smart homes, however, many other systems can be applied.

Our experiments with the fuzzyDL reasoner provide compelling evidence that the algorithm functions correctly according to the defined problem, even as more rules are added to the system. As an additional step towards the validation and applicability of our system, we

**Table 2. Defuzzification and consistency: Experiment two.**

| We codify rules (3), (4), and (5) of the IoT system section |
|---|
| (define-concept R1 (g-and (some light MucLight)(some move MucMove)(some sound MucSound))) |
| (define-concept R2 (implies R1 (some call OnCall))) |
| (define-concept R3 (g-and (some light LowLight)(some move SomMove)(some sound FewSound))) |
| (define-concept R4 (implies R3 (some alarm MedAlar))) |
| (define-concept R5 (g-and (some light LowLight)(not (some move MucMove))(some sound FewSound))) |
| (define-concept R6 (implies R5 (some alarm LowAlar))) |
| |
| (define-concept R7 (and R2 R4)) |
| (define-concept R8 (and R2 R4 R6)) |

| Input | | | | Output | |
|---|---|---|---|---|---|
| Light | Movement | Sound | Query | Output-System | Description |
| | | | Sat | sat subsumes R7? $< = 1.0$ | The rules are consistent 14:21:43:52 F.T. 14:21:45:29. |
| | | | Sat | sat subsumes R8? $< = 0.0$ | The rules are inconsistent The inconsistency is given by Rules R4 and R6. 14:24:30:49 F.T. 14:24:50:97. |

plan to conduct evaluations in real-world scenarios. This will involve implementing our system in smart home environments and collecting real-world usage data. By doing so, we will be able to measure the performance and effectiveness of our system in real-world situations and fine-tune it as needed. This real-world evaluation will also allow us to gather feedback from end users, helping us further tailor the system to meet their needs and ensure practical utility. Additionally, we are open to collaborations with the research community and industry to test and validate our system in various applications and scenarios. These additional steps will strengthen our research's practicality and real-world relevance while fostering collaboration and external validation.

In future work, we plan to explore how the methodology presented in this study can be applied to other IoT systems with varying logic types. Furthermore, we intend to enhance the user interface based on feedback from non-technical users and implement security measures to prevent unintended consequences resulting from user-defined rules.

The system is currently in a prototype stage. The main scalability challenge relies on the fuzzy logic reasoner tool. Other research perspectives include developing and optimizing reasoning algorithms for fuzzy description logic. Similarly, the accuracy of translating natural language instructions into fuzzy logic formulae depends on the accuracy of the corresponding NLP algorithms. We also want to develop NLP interfaces for the proposed frameworks using large language models.

## Author Contributions

**Conceptualization:** Miguel Pérez-Gaspar, Javier Gomez, Everardo Bárcenas, Francisco Garcia.

**Formal analysis:** Miguel Pérez-Gaspar, Javier Gomez, Everardo Bárcenas, Francisco Garcia.

**Investigation:** Miguel Pérez-Gaspar, Javier Gomez, Everardo Bárcenas, Francisco Garcia.

**Software:** Miguel Pérez-Gaspar.

**Writing – original draft:** Miguel Pérez-Gaspar, Javier Gomez, Everardo Bárcenas, Francisco Garcia.

**Writing – review & editing:** Miguel Pérez-Gaspar, Javier Gomez, Everardo Bárcenas, Francisco Garcia.

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
