## [Decision Letter · Decision Letter 0]

29 Aug 2023

PONE-D-23-26012Fuzzy description logics verification on IoT systemsPLOS ONE

Dear Dr. Barcenas,

Thank you for submitting your manuscript to PLOS ONE. After careful consideration, we feel that it has merit but does not fully meet PLOS ONE’s publication criteria as it currently stands. Therefore, we invite you to submit a revised version of the manuscript that addresses the points raised during the review process. Please submit your revised manuscript by Oct 13 2023 11:59PM. If you will need more time than this to complete your revisions, please reply to this message or contact the journal office at plosone@plos.org. Please include the following items when submitting your revised manuscript:A rebuttal letter that responds to each point raised by the academic editor and reviewer(s). You should upload this letter as a separate file labeled 'Response to Reviewers'.A marked-up copy of your manuscript that highlights changes made to the original version. You should upload this as a separate file labeled 'Revised Manuscript with Track Changes'.An unmarked version of your revised paper without tracked changes. You should upload this as a separate file labeled 'Manuscript'.

We look forward to receiving your revised manuscript.

Kind regards,

Nadeem Sarwar

Academic Editor

PLOS ONE

Journal Requirements:

   "This work was supported by the Postdoctoral Scholarship Program at UNAM (Programa de Becas Posdoctorales en la UNAM). The Authors thanks to the UNAM-PAPIIT program IA104122, IA105420, IA102822, CONAHCyT CF 0320403."

   "JG, EB, FG: IA104122, IA105420, IA102822; UNAM-PAPIIT program.

JG: 0320403; Ciencia de Frontera CONAHCyT.

Reviewers' comments:

Reviewer's Responses to Questions

**Comments to the Author**

1. Is the manuscript technically sound, and do the data support the conclusions?

Reviewer #1: Yes

Reviewer #2: Partly

Reviewer #3: Partly

Reviewer #4: Partly

Reviewer #5: Yes

2. Has the statistical analysis been performed appropriately and rigorously? 

Reviewer #1: Yes

Reviewer #2: Yes

Reviewer #3: No

Reviewer #4: No

Reviewer #5: Yes

3. Have the authors made all data underlying the findings in their manuscript fully available?

Reviewer #1: Yes

Reviewer #2: Yes

Reviewer #3: No

Reviewer #4: No

Reviewer #5: Yes

4. Is the manuscript presented in an intelligible fashion and written in standard English?

Reviewer #1: Yes

Reviewer #2: Yes

Reviewer #3: Yes

Reviewer #4: Yes

Reviewer #5: Yes

5. Review Comments to the Author

Reviewer #1: 1- The introduction section was written well and included in clarifying the basics of the research field of the submitted scientific paper, but no reference was made to the paragraphs it contains of information or conclusions

2- The references in the Related work section are arranged from oldest to newest, except for reference 27. In addition, the researcher did not show weaknesses in each reference compared to his contributions to this research presented

3-The researcher did not discuss the difference between the proposed method and previous methods that were applied to demonstrate the efficiency of the system compared to other previous systems. Also, the researcher did not provide a future vision for the work submitted to benefit from it by the researchers in building a future vision in the field of work.

4-Table 2 is not in the right place

5-Standardization of reference format. In addition to correcting the year of publication for reference 35

Reviewer #2: The authors have done very interesting work and idea. They may incorporate the following suggestion

Bring the numerical results in the abstract and some future suggestions in the last line.

Revise the last para in the Introduction as it is machine generated.

The para starting from 170 may be revised.

The fuzzy logic rules need explanation in proper and simple way.

Experiment portion needs more explanation and at the conclusion some future directions may also be drawn.

Include some latest references to relate the work to state of the art.

Reviewer #3: 1.How does the proposed framework ensure that users with no technical knowledge can effectively utilize it to build their own IoT applications? Are there any potential challenges or limitations in terms of the learning curve or usability for non-technical users?

2.Can the authors provide more details on the specific implementation and integration of the friendly interface block? How does it handle complex scenarios or conditions that may require multiple rules or dependencies?

3.What are the specific limitations of using fuzzy logic reasoning in the proposed framework? Are there any potential drawbacks or trade-offs associated with relying on fuzzy logic for translating human language to specific actions?

4.How does the proposed system detect and handle inconsistencies in real-time? Are there any potential false positives or false negatives in the detection process? How does it ensure the accuracy and reliability of the detection mechanism?

5.Have the authors considered the scalability and performance implications of the framework when applied to larger and more complex IoT applications? How does it handle increasing rule complexity or a larger number of connected devices?

6.What are the security considerations and measures implemented in the proposed smart-home IoT security system? How does the framework address potential vulnerabilities or threats associated with user-defined rules and actions?

7.Have the authors conducted user studies or evaluations to validate the usability and effectiveness of the framework for non-technical users? How do they ensure that the system meets the diverse needs and expectations of different end-users?

8.How does the proposed framework handle maintenance and updates of the IoT applications built by users? Are there mechanisms in place to address evolving requirements or changes in the IoT ecosystem?

9.Can the authors provide a comparison or discussion on how their framework differs from existing solutions that aim to empower end-users in building their own IoT applications? What are the unique contributions or advantages of the proposed framework in this context?

Reviewer #4: The paper title is vague. As mentioned in the abstract, the paper presents a framework. However, the title of the paper does not reflect this. The search title needs to be modified.

It is difficult to verity the validity of the proposed model through examples. More experiments and comparisons with existing state-of-the-art approaches are need (in terms of percentage of data errors and complexity.

Reviewer #5: Good day. Your submission was well written but lacks a few things. You will find attached some comments made regarding your submission. Please ensure you make all the necessary corrections before resubmitting.

6. PLOS authors have the option to publish the peer review history of their article (what does this mean?). If published, this will include your full peer review and any attached files.

Reviewer #1: No

Reviewer #2: **Yes: **Naveed Abbas

Reviewer #3: **Yes: **Tarek Abd El-Hafeez

Reviewer #4: No

Reviewer #5: No

---

## [Author Response · Author response to Decision Letter 0]

20 Oct 2023

It is a pleasure to respond to the valuable comments and suggestions provided by the editor and the reviewers regarding our manuscript titled "Fuzzy description logics verification on IoT systems." We sincerely appreciate the time and effort dedicated to reviewing our work, which has undoubtedly contributed to improving the quality and robustness of our research.

We have carefully considered each of the comments and suggestions raised in the reviews. Below, we provide our responses and the modifications made to the manuscript to address each point specifically:

Reviewer 1:

Comment:

The introduction section was written well and included in clarifying the basics of the research field of the submitted scientific paper, but no reference was made to the paragraphs it contains of information or conclusions. 

Response

Based on the reviewer's observation, we added some references in the Introduction section ([1] and [7]).

Comment:

The references in the Related work section are arranged from oldest to newest, except for reference 27. In addition, the researcher did not show weaknesses in each reference compared to his contributions to this research presented.

Response:

Based on the reviewer's observation, we added the weaknesses of the references in the Related Work Section. These changes can be seen in lines 103-107, 114-116, 128-129, and 131-132, and we also summarize the weakness of the related work and the advantages of the proposed model in lines 133-147.

Comment:

The researcher did not discuss the difference between the proposed method and previous methods that were applied to demonstrate the efficiency of the system compared to other previous systems. Also, the researcher did not provide a future vision for the work submitted to benefit from it by the researchers in building a future vision in the field of work

Response:

At the end of the related work section, we added a paragraph discussing the main differences between related works in the literature and our work (lines 133-147). Similarly, we added future directions of this work at the end of the conclusions (lines 549-558).

Comment:

Table 2 is not in the right place. Standardization of reference format. In addition to correcting the year of publication for reference 35

Response:

Thanks for pointing this out; The location of Table 2 has been corrected (page 18), and the reference format has been standardized, including the correction of the publication year for reference 35 (now it is 9).

Reviewer 2

Comment:

The authors have done very interesting work and idea. They may incorporate the following suggestion Bring the numerical results in the abstract and some future suggestions in the last line.

Response:

Results on the formal description of IoT consistency and experiments are now included in the abstract. Further research perspectives are also incorporated in the Conclusion Section.

Comment:

Revise the last para in the Introduction as it is machine generated. The para starting from 170 may be revised.

Response:

In response to concerns about the text's authenticity in this era of AI, it is important to emphasize that the researchers have authored all the text in the article.

The modification of the Introduction is as follows (line 83): The structure of this work is organized as follows: In the first section, we discuss previous research to provide context for our study. Then, in the IoT Systems section, we define the system and introduce the concept of consistency. Next, in the Fuzzy Description Logic Verification section, we explain the basics of fuzzy logic and present a result that shows how consistency relates to both the IoT system and fuzzy description logic. Ultimately, in the Fuzzy Control for an IoT Security System section, we outline the next steps, which include providing context, setting system rules, and putting the system into action. Finally, we present the program’s syntax and share the experiments we conducted to support and illustrate the concepts we’ve discussed. The modification of paragraph 170 is as follows (line 197): Recall the example of the sound sensor intervals intersecting: if the sensed value falls within this intersection, our system employs a special structure to determine which interval is closer to the sensed value.

Comment:

The fuzzy logic rules need explanation in a proper and simple way.

Response:

We now include an extensive explanation, with more new examples, of these rules in the corresponding section.

Comment:

Experiment portion needs more explanation and at the conclusion some future directions may also be drawn.

Response:

The corrections have been made appropriately, following the reviewer's recommendations. The explanations in the experiments section have been expanded to lines 500-525, and ideas for future research directions have been added (lines 549-558).

Reviewer 3

Comment:

How does the proposed framework ensure that users with no technical knowledge can effectively utilize it to build their own IoT applications? Are there any potential challenges or limitations in terms of the learning curve or usability for nontechnical users? 

Response:

Regarding users with no technical knowledge, the framework can process simple instructions in the form “if … then …”. A prior configuration of the framework is required to define an execution context; for instance, what does it mean “much noise”. This configuration can be done by experts or users with more experience. This observation is explained in detail in the fuzzy description logic verification section.

Comment:

Can the authors provide more details on the specific implementation and integration of the friendly interface block? How does it handle complex scenarios or conditions that may require multiple rules or dependencies?

Response:

Ideally, as long as the user is able to input the rules to the system with the correct syntax (consistent or not), no matter how complex the scenario is, the system should operate (either by doing what the rules tell the system to do, or showing a no consistency alert that prompts the user to take further actions).

In this work, we focus on systems where non-technical users interact with the system; thus, we anticipate the system will perform simple tasks, and thus few rules will be involved. However, while in theory, the FuzzyFL reasoner we used in this work has no known limit in the number of rules (or their complexity) it can handle, in practice, the hardware (CPU and memory) of the computer the system runs does impact the system’s response time. 

We added some description about the computer we used and the response times of the shown experiments to highlight this point in the new manuscript. 

It remains for future work to test the limits (if any) of the reasoner (fuzzyDL) as we scale up the number of rules and their impact on the whole system.

Comment:

What are the specific limitations of using fuzzy logic reasoning in the proposed framework? Are there any potential drawbacks or trade-offs associated with relying on fuzzy logic for translating human language to specific actions?

Response:

Accuracy in translating natural language instructions in terms of fuzzy logic formulae depends on the accuracy of the corresponding NLP algorithms. We clarify this observation in the conclusions in lines 553-558.

Comment:

How does the proposed system detect and handle inconsistencies in real-time? Are there any potential false positives or false negatives in the detection process? How does it ensure the accuracy and reliability of the detection mechanism?

Response:

Inconsistencies are detected before the execution of the system. Theorem 1 provides a mathematical guarantee that the system is free of inconsistencies under any real-time scenario. This is clarified before the statement of Theorem 1 in lines 327-331.

Comment:

Have the authors considered the scalability and performance implications of the framework when applied to larger and more complex IoT applications? How does it handle increasing rule complexity or a larger number of connected devices?

Response:

The system is currently in the prototype stage. The main scalability challenge relies on the fuzzy logic reasoner tool. This is explained in the conclusions in lines 553-558.

Comment:

What are the security considerations and measures implemented in the proposed smart-home IoT security system? How does the framework address potential vulnerabilities or threats associated with user-defined rules and actions?

Response:

The current work primarily focussed on letting the users input rules, process those rules, and verify the consistency of the whole system. In this respect, providing security measures to avoid unintentional harm by user-defined rules was out of the scope of this paper. 

However, we recognize the relevance of security in the system and plan to study it as part of future work. We actually believe there is space in the logic (fuzzy or not) to evaluate such potential harm and quantify it numerically in a similar way consistency is quantified. We incorporate this point as part of future work in the conclusions.

Comment:

Have the authors conducted user studies or evaluations to validate the usability and effectiveness of the framework for non-technical users? How do they ensure that the system meets the diverse needs and expectations of different end-users?

Response:

We let a few users, apart from the authors, interact with the interface to verify its basic operation. However, we never conducted a deep study of its usability considering a wide pool of people with different backgrounds. However, this is something we plan to look at in the future and it is mentioned at the end of the conclusions.

Comment:

How does the proposed framework handle maintenance and updates of the IoT applications built by users? Are there mechanisms in place to address evolving requirements or changes in the IoT ecosystem?

Response:

The current framework did not consider maintenance or updates, as the main focus of the work was to validate the consistency of user-inserted rules. However, we anticipate the persons maintaining the framework (app) will perform these duties periodically, similar to updating any other app in a mobile phone ecosystem, for example. 

Comment:

Can the authors provide a comparison or discussion on how their framework differs from existing solutions that aim to empower end-users in building their own IoT applications? What are the unique contributions or advantages of the proposed framework in this context?

Response:

We added a whole paragraph at the end of the related work section (lines 133-147), comparing our work with other related proposals and highlighting the key differences.

Reviewer 4

Comment:

The paper title is vague. As mentioned in the abstract, the paper presents a framework. However, the title of the paper does not reflect this. The search title needs to be modified.

Response:

The title has been updated to 'A fuzzy description logic based IoT framework: verification and end-user programming' in response to your feedback. We believe this new title accurately represents our work and appreciate your input in improving our paper.

Comment:

It is difficult to verify the validity of the proposed model through examples. More experiments and comparisons with existing state-of-the-art approaches are needed (in terms of percentage of data errors and complexity.

Response:

The validity of the proposed model is formally stated and proved in Theorem 1. 

Reviewer 5

Comment:

In the abstract (line 11), the word built should be changed to build (to build their own IoT applications).

Response:

The abstract correction has been made as requested (line 7).

Comment:

The headings and subheadings are not numbered/lettered.

Response:

Headings and subheadings are not numbered per the guidelines.

Comment:

The paragraph (lines 46-53) has no citation. Also, lines 54-59. Cite appropriately.

Response:

We have thoroughly reviewed our manuscript and have identified the relevant sources to support the information presented in those paragraphs (line 57).

Comment:

Fig 2, Fig 4, and Fig 5 are blurry on some parts. Clearer images should be uploaded with better resolution.

Response:

All figures in the document have been corrected, and clearer images with improved resolution have been uploaded

Comment:

In the section of Actuator system (line 377), the statement is ambiguous. Kindly paraphrase.

Response:

The requested modification has been made in the "Actuator system" section. The revised sentence now reads (lines 431-433).

Comment:

The performance and efficiency of the fuzzy logic reasoner blocks in real-world applications have not been outlined in the paper.

Response:

We now include results on performance in Table 1 (page 17) and Table 2 (page 18).

Comment:

Only a smart-home IoT security system was considered in the paper. As such, it is uncertain how the framework would function when used for other applications.

Response:

Because of size limitations, we have to constrain the testing to only the smart-home IoT security system discussed in the paper. However, we anticipate the proposed framework can be applied to any other applications as long as the new system can separate the task responsibility of the expert (ABox) to the task concerning the user (TBox).

Comment:

There is no performance evaluation of the framework. Having this will enhance the quality of the paper.

Response:

We have extended the experiments section to include performance evaluation.

We appreciate once again the opportunity to submit our work for expert review, and we are confident that the revisions made have significantly strengthened the quality and value of our article. We hope that the additional revisions reflect these improvements and lead to the eventual publication of our work in PLOS ONE.

We remain at your disposal for any further comments or clarifications that may be necessary.

Sincerely,

The authors

---

## [Editor Report · Decision Letter 1]

31 Oct 2023

PONE-D-23-26012R1A fuzzy description logic based IoT framework: formal verification and end user programmingPLOS ONE

Dear Dr. Barcenas,

Thank you for submitting your manuscript to PLOS ONE. After careful consideration, we feel that it has merit but does not fully meet PLOS ONE’s publication criteria as it currently stands. Therefore, we invite you to submit a revised version of the manuscript that addresses the points raised during the review process.

We look forward to receiving your revised manuscript.

Kind regards,

Nadeem Sarwar

Academic Editor

PLOS ONE

---

## [Author Response · Author response to Decision Letter 1]

16 Nov 2023

It is a pleasure to respond to the valuable comments and suggestions provided by the editor and the reviewers regarding our manuscript titled "Fuzzy description logics verification on IoT systems." We sincerely appreciate the time and effort dedicated to reviewing our work, which has undoubtedly contributed to improving the quality and robustness of our research.

We have carefully considered each of the comments and suggestions raised in the reviews. Please find a detailed response to each comment, in the corresponding attached document "Response to Reviewers".

---

## [Decision Letter · Decision Letter 2]

18 Dec 2023

A fuzzy description logic based IoT framework: formal verification and end user programming

PONE-D-23-26012R2

Dear Dr. Barcenas,

We’re pleased to inform you that your manuscript has been judged scientifically suitable for publication and will be formally accepted for publication once it meets all outstanding technical requirements.

Kind regards,

Nadeem Sarwar

Academic Editor

PLOS ONE

Additional Editor Comments (optional):

Reviewers' comments:

Reviewer's Responses to Questions

**Comments to the Author**

1. If the authors have adequately addressed your comments raised in a previous round of review and you feel that this manuscript is now acceptable for publication, you may indicate that here to bypass the “Comments to the Author” section, enter your conflict of interest statement in the “Confidential to Editor” section, and submit your "Accept" recommendation.

Reviewer #1: All comments have been addressed

Reviewer #3: All comments have been addressed

2. Is the manuscript technically sound, and do the data support the conclusions?

Reviewer #1: Yes

Reviewer #3: Partly

3. Has the statistical analysis been performed appropriately and rigorously? 

Reviewer #1: Yes

Reviewer #3: Yes

4. Have the authors made all data underlying the findings in their manuscript fully available?

Reviewer #1: Yes

Reviewer #3: Yes

5. Is the manuscript presented in an intelligible fashion and written in standard English?

Reviewer #1: Yes

Reviewer #3: Yes

6. Review Comments to the Author

Reviewer #1: (No Response)

Reviewer #3: All comments were thoroughly reviewed and addressed through appropriate changes to the text. I believe the manuscript is now suitable for publication.

7. PLOS authors have the option to publish the peer review history of their article (what does this mean?). If published, this will include your full peer review and any attached files.

Reviewer #1: No

Reviewer #3: **Yes: **Tarek Abd El-Hafeez

---

## [Editor Report · Acceptance letter]

3 Mar 2024

PONE-D-23-26012R2 

PLOS ONE

Dear Dr. Bárcenas, 

I'm pleased to inform you that your manuscript has been deemed suitable for publication in PLOS ONE. Congratulations! Your manuscript is now being handed over to our production team.

Kind regards, 

on behalf of

Dr. Nadeem Sarwar 

Academic Editor

PLOS ONE